# Well-defined nanostructuring with designable anodic aluminum oxide template

Rui Xu[1], Zhiqiang Zeng[1] & Yong Lei [1]✉

Well-defined nanostructuring over size, shape, spatial configuration, and multi-combination is a feasible concept to reach unique properties of nanostructure arrays, while satisfying such broad and stringent requirements with conventional techniques is challenging. Here, we report designable anodic aluminium oxide templates to address this challenge by achieving well-defined pore features within templates in terms of in-plane and out-of-plane shape, size, spatial configuration, and pore combination. The structural designability of template pores arises from designing of unequal aluminium anodization rates at different anodization voltages, and further relies on a systematic blueprint guiding pore diversification. Starting from the designable templates, we realize a series of nanostructures that inherit equal structural controllability relative to their template counterparts. Proof-of-concept applications based on such nanostructures demonstrate boosted performance. In light of the broad selectivity and high controllability, designable templates will provide a useful platform for well-defined nanostructuring.

[1] Fachgebiet Angewandte Nanophysik, Institut für Physik & IMN MacroNano, Technische Universität Ilmenau, Ilmenau 98693, Germany.
✉email: yong.lei@tu-ilmenau.de

Due to the compelling requirement of device miniaturiza-tion, synthesis of nanoscopic structures and their macroscopic integration into a large-scale array are fun-damental to modern and future devices in the fields of optics[1], electronics[2], telecommunication[3], biology[4], energy conversion/storage[5,6], and stimuli-responsive materials[7], etc. It is known that nanostructures are subject to physical and chemical property variation as a function of their geometry and composition;[8] and arrayed assemblies of these nanostructures exhibit collective behaviors of their responses in terms of coupling in the same set and synergy between different sets[9,10]. Therefore, to tailor the overall properties of a nanostructure array (hence to foster devices based on this nano-array), it is highly desirable to achieve well-defined nanostructuring that is capable of precise controlling over the structural parameters of a nanostructure array. As such, six capabilities could be indispensable for an efficient well-defined nanostructuring technique: i) ability to assemble nanostructures into a large-scale array with cost-effective processes; ii) reliable size controllability as well as iii) in-plane and iv) out-of-plane shape designability of the nanostructures; v) alterability of the spatial configuration of nano-arrays; vi) compatibility of different sets of arrayed nanostructures with tunable shape and config-uration of each set. Various nanostructuring techniques have been developed, such as photo/electron-beam lithography, self-assembly, nanoimprinting, and template-based techniques[11] as well as material growth controlling[12], while almost none of these techniques fulfills all the above six capabilities of well-defined nanostructuring.

The template-based technique, especially of using anodic alu-minum oxide (AAO) nanoporous template, has attracted high attention for nanostructuring[13], because it fully satisfies the aforementioned first and second capabilities: i) enabling inte-gration of millions of nanostructures into large-scale arrays in a cost-effective way;[14] ii) AAO pores along with the replicated nanostructure arrays are well controlled in size[15,16]. However, other four capabilities (iii–vi) are missing for AAO templates. Different from self-ordered two-step anodic anodization which only generates circular-shaped pores with close-packed honey-comb (i.e., trigonal) arrangement, artificially nanoengineering Al-foil surface by hard imprint lithography can guide anodic ano-dization to form pores with desired structural parameters (e.g., arrangement and interpore spacing) that highly depend on the surface topography of imprinting stamps[17]. To date, only a handful of in-plane shapes are available for pores (e.g., circular, triangular, square, and diamond)[18–21]. Although the out-of-plane pore size is tunable along the pore axis (e.g., by tuning anodiza-tion current or voltage), their constituent in-plane shapes remain the same from top to bottom (i.e., circular shape with different diameters)[22,23]. The spatial configurations of pores are quite limited (e.g., the trigonal, tetragonal, hexagonal arrangements, and their combinations with identical interpore spacing)[18–21], which are governed by the linear relationship between pore spacing and AV[24] and greatly restrict pore-shape diversification considering the determinative dependence of pore shape on spatial configuration[19]. Binary-pore AAO templates enable two sets of nanostructures into one matrix[25], however only few combinations are achievable due to the poor selectivity in the pore shape (i.e., the 1st-set anodized pores only have an square shape) and spatial configuration (i.e., all sets of pores are restricted to a tetragonal arrangement). Therefore, to realize well-defined nanostructuring based on template-based techniques, breakthroughs must be made from the functional limitation of the existing AAO templates, especially to endow the templates with both broad selectivity (in pore shape, pore spatial config-uration, and pore combination) and high controllability (in each parameter).

Here, we realize designable AAO templates with state-of-the-art controllability over in-plane and out-of-plane pore shape, spatial configuration of pore arrangement, and pore combination. By intentionally introducing unequal aluminum anodization rate and further modulating anodization rate difference at different AVs, the in-plane pore shape of the designable template can be continuously altered from polygons (e.g., triangle and square) with internally-bent (i.e., concave) walls to polygons with non-bent walls (i.e., straight) and then to polygons with externally-bent (i.e., convex) walls (please see Supplementary Fig. 1 for the schematic illustration of shape-different pores) as compared to a conventional template with a single pore shape in a specific spatial configuration. These pore shapes can be integrated into one pore along the axial direction by sequential anodization at different AVs, forming out-of-plane multi-segment pores with different shapes (not only sizes) for every segment. The desig-nable templates are also capable of mixing diverse spacing-different pores into one matrix, successfully pushing the selection in spatial configuration beyond the aforementioned identical spacing limitations. Furthermore, a large series of different-set pore combinations, where each set of pores possesses designable in-plane/out-of-plane shape and spatial configuration, are con-structed. Importantly, the structural controllability of the desig-nable templates is totally transferable into their nanostructure counterparts, achieving a large quantity of (size-, shape-, spatial configuration-, and combination-) well-defined nanostructures (nanoparticles, nanotubes, nanowires, and nanomeshes) which enable device performance optimization that is demonstrated in three proof-of-concept applications.

## Results

**Designing principle of in-plane pore shape.** The in-plane pore shape designability of AAO templates originates with a scenario of designing unequal aluminum anodization rates at different AVs. Here, we select potentiostatic anodization for structural controlling of pores in consideration of the linear relationship between AV and pore parameters (e.g., interpore distance and pore diameter)[17]. Given that nanoimprinting Al-foil surface with appropriate texture could guide the initiation of pores[19], we hope to generate unequal aluminum anodization rates by introducing uneven-profiled four-leaf clover-like nanoconcaves onto surface (see layout of COMSOL simulation in Supplementary Fig. 2). Fabrication process of a proof-of-principle template is schema-tically depicted in Fig. 1c, which generally includes two sequential procedures of imprinting and anodization. Previous reports point out that plastic flow is damped in thin oxide films[26] and that the electric-field-assisted oxide dissolution is primarily responsible for the formation of incipient pores[27]. Therefore, we simulate electric field (EF) maps at the early stage of anodization to elu-cidate how one can control pore shape. Here, an arbitrary spacing of 400 nm is set for the preset nanoconcave array. Following the linear spacing-AV relation, the appropriate AV should be 160 V and accordingly, aqueous solution of 0.4 M $H_3PO_4$ is used as anodization electrolyte to form nanoporous structures[24]. Obviously, high EF sites are located at the bottom of nano-concaves (Fig. 1a) and eight spots on the walls (Fig. 1b). According to the field-assisted dissolution theory in which alu-minum anodization is preferably conducted at high EF sites[17], here anodization will proceed not only at the bottoms of nano-concaves but also on the walls, resulting in axial anodization imposing on pore elongation and radial anodization dictating shape evolution. As shown in Fig. 1b, apart from eight stronger-EF spots, each nanoconcave has weaker EF at four humps, leading to uneven EF distribution and consequently unequal radial ano-dization rates. As previously reported, acid anions are driven into

**Fig. 1 Fabrication of templates with in-plane shape-designable pores.** COMSOL-simulated electric fields at **a** vertical cross-section and **b** near-surface lateral cross-section of an aluminum foil patterned with four-leaf clover-like nanoconcaves of 400-nm spacing and tetragonal arrangement under an anodization voltage (AV) of 120 V, the dashed line in **a** at the half-depth of nanoconcaves marks the near-surface lateral cross-section in **b**. **c** Schematic illustration for the fabrication process of AAO template (from left to right): fabricating Ni imprint stamp and electro-polishing aluminum foil; transferring the structural feature of Ni stamp by imprinting to equip aluminum surface with an array of four-leaf clover-like nanoconcaves; anodizing the imprinted area at different AVs. Scale bars: 200 nm.

the oxide layer during anodization, and the number of incorporated acid anions is positively related to the volume expansion[17,28]. Therefore, more anions in the electrolyte ($PO_4^{3-}$) will be incorporated into the eight spots because of the higher electric fields (Fig. 1b), leading to larger volume expansion than that of the four humps. With the AV increasing, the absolute EF difference between eight spots and four humps gets larger (Supplementary Fig. 3), which consequently causes larger volume expansion difference. Based on these results, it is predicted that larger volume expansion difference at higher AVs should lead to smoother and more externally-bent walls, vice versa sharper and more internally-bent ones at lower AVs (please refer to Supplementary Fig. 4 for pore shape evolution). Because stress starts to generate with the formation of oxide[29], especially considering uneven anion incorporation (and volume expansion) in our case, it is believed that the electric-field-assisted dissolution and the stress-driven oxide flow work together to prolong the pores at the steady state of anodization[19,30]. This prediction of AV-dependent pore shape designability is fully confirmed by the following experimental results.

**Designing in-plane pore shape by adjusting anodization voltage.** A purposely-designed Ni imprint stamp decorated with four-leaf clover-like nanopillars on its surface (Supplementary Fig. 5) was used for predetermining sites on the aluminum surface (Supplementary Fig. 6). Anodization was then performed with four different AVs: at 120 V, the anodized pores were cross-shaped with internally-bent walls (Fig. 2a and Supplementary Fig. 7a); with 140 V, the internally-bent amplitude was suppressed to form star-like pores (Fig. 2b); at 160 V, the pore wall became straight without bending, achieving square-shaped pores (Fig. 2c); when the AV was 200 V, the pores presented a circular shape, analogous to a square shape with its walls being externally-bent (Fig. 2d and Supplementary Fig. 7b). To discern the crucial role of inhomogeneous radial anodization rates (i.e., uneven EF distribution) in shape designability, we fabricated circular nanoconcaves for reference by using a Ni circular-pillar stamp (Supplementary Fig. 8) and achieved homogeneous radial anodization rate (as evidenced by even EF distribution) on the sidewalls of nanoconcaves (Supplementary Fig. 9d). Not surprisingly, the pore shape cannot be altered no matter what AV was applied (Supplementary Fig. 10).

To test the applicability of the pore shape designability to other arrangements, we also made a hexagonal array of three-leaf clover-like nanoconcaves (Supplementary Figs. 11, 12). It was found that also guided by uneven EF distribution (Supplementary Fig. 13d), the anodized pores possessed a similar shape-designing trend to that of the tetragonal array. For example, the triangular pore shape without bending obtained at 140 V (Fig. 2g and

Supplementary Fig. 14a) was changed to a wall-internally-bent triangular shape at a lower AV of 120 V (Fig. 2f) and further evolved to a more wall-internally-bent triangular shape when the AV was decreased to 100 V (Fig. 2e and Supplementary Fig. 14b); when the AV was increased to 155 V, it demonstrated a wall-externally-bent shape (Fig. 2h). To explore appropriate AV values, we performed anodization in a broader AV range thereafter. It is found that concerning a specific arrangement, the adjustable AV for designing pore shape is limited in an AV range (denoted as appropriate AV range), out of which (i.e., in too-low AV and too-high AV ranges) the arrangement predetermined by nanoconcaves is broken (Supplementary Fig. 15). The three AV ranges are separated by two threshold values (Supplementary Fig. 16), which are empirically observed to be $V_a$ and $\sqrt{3}V_a$, where $V_a = 0.4$ (V nm$^{-1}$) $\times L_h$ (nm) and $L_h$ is the interpore spacing of the hexagonal array (i.e., $400/\sqrt{3}$ nm). In other words, AV thresholds can be derived from the linear spacing-AV relation, i.e., AV (V) = spacing (nm) $\times 0.4$ (V nm$^{-1}$)[24], regarding two spacings (e.g., $L_h$ and $\sqrt{3}L_h$) of emerging arrays. This in-plane shape-designable technique is applicable for large-scale fabrication, as evidenced by a template with a 2.5-cm-diameter area (Supplementary Fig. 17). The green color, spreading over the whole anodized area (left image of Fig. 2i), implies high structural uniformity of the pore array which is verified by a large-area 40-μm-width scanning electron microscope (SEM) image (right part of Fig. 2i) and a nearly monodisperse pore size distribution (Supplementary Fig. 18). Here, for each pore arrangement, we just selected four AVs and obtained four pore shapes. Of course, the rest of values within the appropriate AV range are selectable for tuning pore shape and consequently, more pore shapes should be achievable. In other words, one stamp can lead to many shape-different templates by easily controlling AV, thus avoiding frequent fabrication of mask/stamp with different shapes in conventional lithographic techniques.

**Designing out-of-plane pore shape with sequential anodization voltages.** To further advance the pore shape designability, we also made efforts toward tuning the out-of-plane shape (i.e., forming multiple segments with distinct shapes for each segment). Using the tetragonal template as an example, Fig. 3 demonstrates how to design pore shape in the axial direction by multi-step anodization with different AVs. For the template I in Fig. 3 was anodized at 120 V, the cross-shaped pores remain identical from top to bottom. When two AVs of 120 and 200 V were sequentially applied to prepare the template II, its pores contain two segments: the first segment at the top has a cross shape while the second at the bottom has a circular shape. Although the first segment was exposed to higher AV during the second-step anodization,

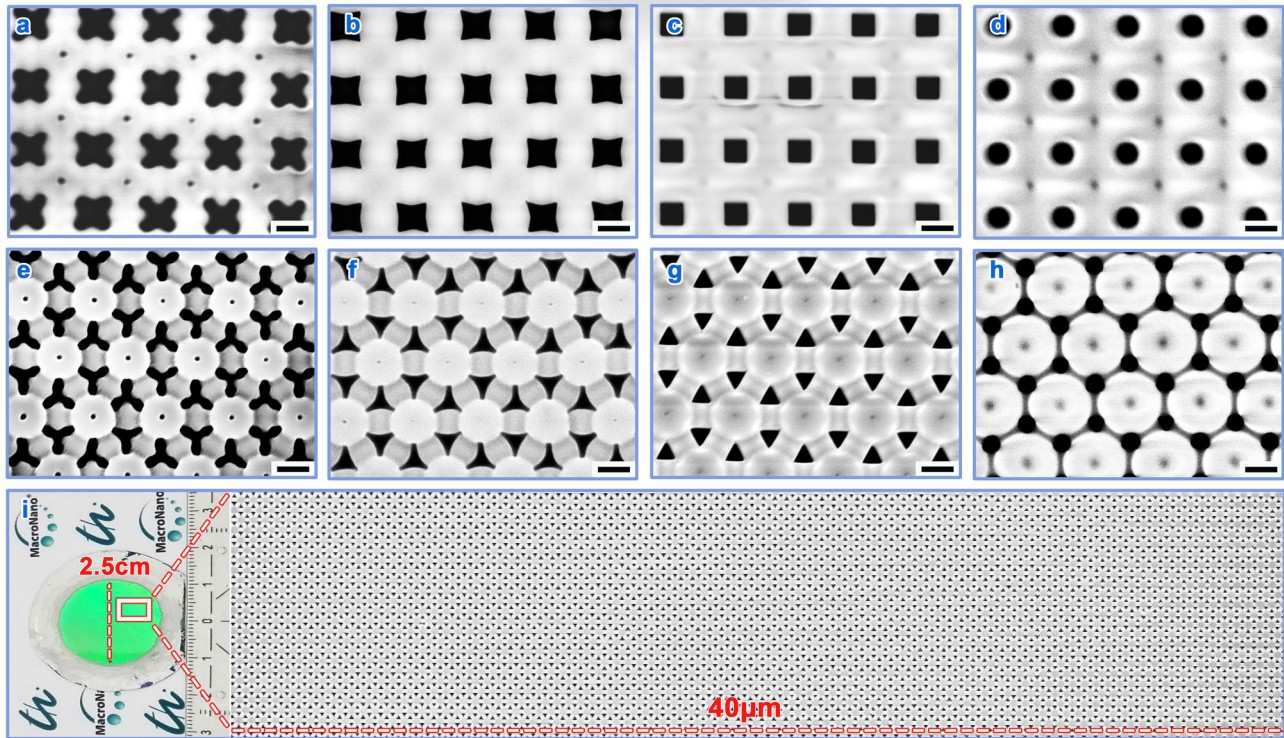

**Fig. 2 Templates with in-plane shape-designable pores.** SEM images of templates with shape-designable pores in **a–d** tetragonal and **e–h** hexagonal arrangements. Two arrangements are 400 and $400/\sqrt{3}$ nm in spacing, respectively (see Supplementary Figs. 5,11). From left to right, the exploited AVs were incrementally increased, with **a** 120 V, **b** 140 V, **c** 160 V, **d** 200 V; and **e** 100 V, **f** 120 V, **g** 140 V, **h** 155 V, respectively. **i** Optical photograph of 2.5-cm-diameter AAO template (left) and large-area SEM image (right). Scale bars: 200 nm.

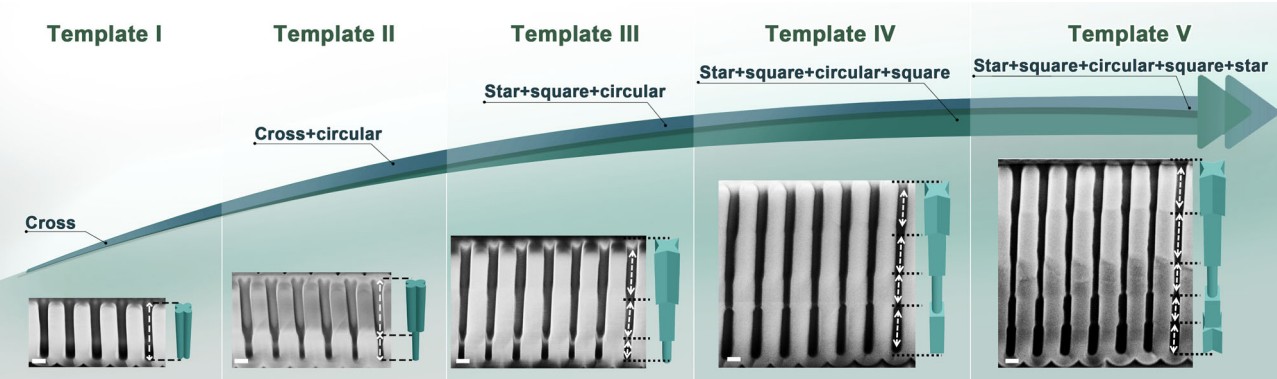

**Fig. 3 Templates with out-of-plane shape-designable pores.** Cross-sectional SEM images and the corresponding pore schematics (on the right of each SEM figure) of templates I–V with different out-of-plane pore shapes. From left to right: the sequentially exploited AVs (and the pore shapes) are Template I: 120 V (cross); Template II: 120→200 V (cross + circular); Template III: 140→160→200 V (star + square + circular); Template IV: 140→160→200→160 V (star + square + circular + square); Template V: 140→160→200→160→140 V (star + square + circular + square + star), respectively. Scale bars: 200 nm.

the cross-pore shape remained unchanged, indicating that the shape in each segment is independent. This independence imparts more diverse axial pore shape designability, as reflected by the template III which has three segments of each pore: star, square, and circular shapes corresponding to the AVs of 140, 160, and 200 V, respectively. In addition to the adjusting trend from low to high AVs, we can also perform multi-step anodization in the opposite direction, for example, to further carry out multi-step anodization on the template IV (from 200 to 160 V) and the template V (from 200 to 160 and then 140 V). Accordingly, template IV with four segments (star + square + circular + square) and template V with five segments (star + square + circular + square + star) were fabricated. It is foreseeable that templates with more selections of out-of-plane pore shapes, as

well as the resultant one-dimensional nanostructures of multiple shape-different segments (see examples in Supplementary Fig. 19, Fig. 6e–h), are achievable by selecting different series of AVs in the appropriate AV range.

**Mixing spacing-different arrangements for designing pore shape.** Given that pore shapes are highly dependent on the spatial configuration of neighboring pores (Supplementary Fig. 20 for details)[19,21] and also considering that only three equilateral polygons tile a plane without gap as well as limited numbers of combination of equilateral polygons (corresponding to the conventional linear spacing-AV relation[24], please refer to Supplementary Fig. 21), it is imperative to mix arrangements of different

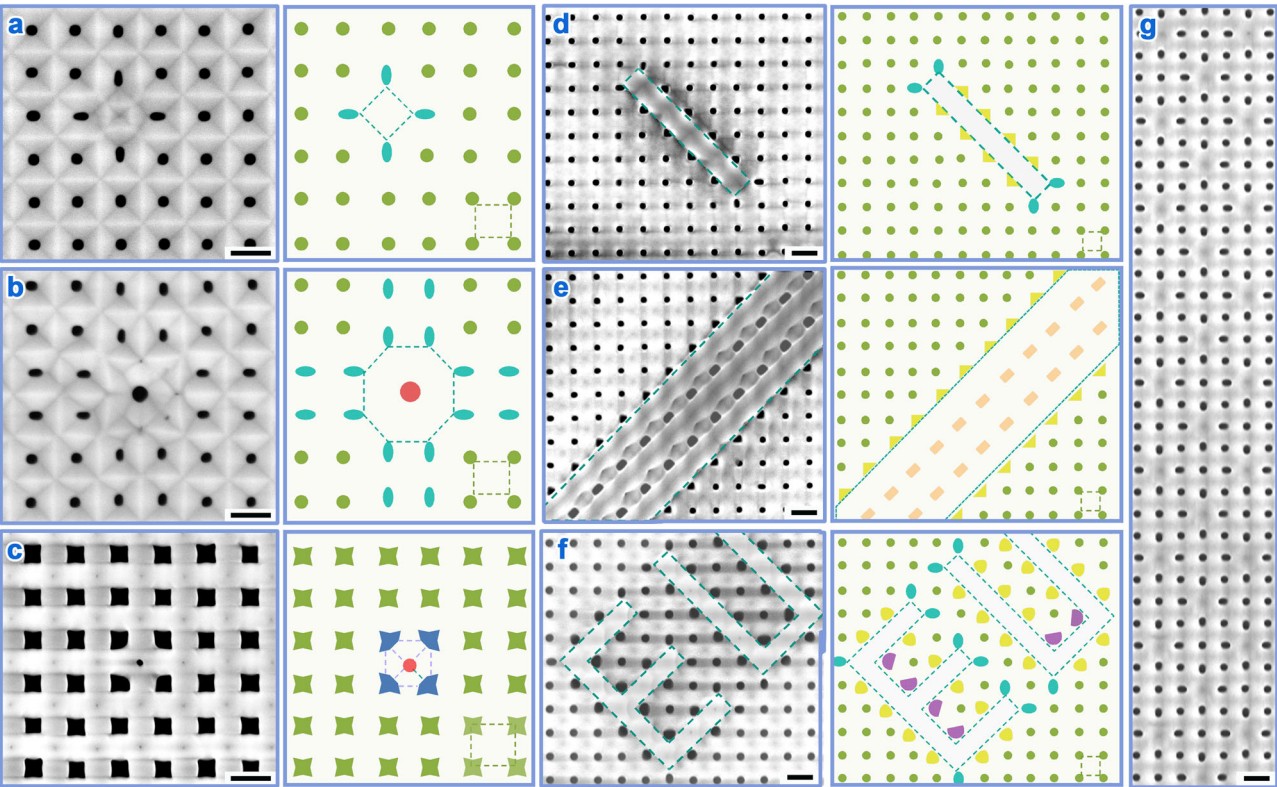

**Fig. 4 Templates with mixture spacing-different arrangements.** SEM images (left) and schematics (right) of basic tetragonal templates mixed with accessory point units of (**a**) larger-spacing tetragonal arrangement, (**b**) octagonal arrangement, and (**c**) centered tetragonal arrangement, as well as (**d**) line, **e** area, and (**f**) pattern "EU" of larger-spacing tetragonal arrangements. The pores in the basic arrangement are exhibited by green color, and those pores in accessory arrangements are by cyan, red, blue, yellow, orange, and purple colors. **g** Large-area SEM image of a mixture of two spacing-different tetragonal arrangements. Scale bars: 400 nm.

spacings for achieving more spatial configurations and thus more shapes of pores. Considering the preceding observation that AVs for a specific arrangement are limited within an appropriate AV range (Supplementary Fig. 16), we hypothesized that AVs of a spacing-different mixture arrangement should lie in an intersection of several appropriate AV ranges to simultaneously prevent the occurrence of new pores and the disappearance of existing pores for every constituent arrangement. To test this hypothesis, we mixed the above tetragonal arrangement with spacing of $L_t$ (i.e., 400 nm) with a larger-spacing ($\sqrt{2}L_t$) tetragonal arrangement (Supplementary Fig. 22, a–c). In theory, the appropriate AV ranges for two arrangements are ($V_b/\sqrt{2}$, $\sqrt{2}V_b$) and ($V_b$, $2V_b$) (see Supplementary Fig. 22, d–f for details), where $V_b = 0.4$ (V nm$^{-1}$) $\times L_t$ (nm)[24]. The intersectional AVs locate from $V_b$ (160 V) to $\sqrt{2}V_b$ (~226 V). When an AV of 190 V within the intersection was exploited, both arrangements were maintained (Fig. 4a); in contrast, the long-spacing arrangement at an AV of 158 V below the intersection was absent with the occurrence of new pores at the central sites (Supplementary Fig. 23). Regarding the octagonal arrangement with an appropriate AV range from $\sqrt{2.5}V_b$ to $\sqrt{5}V_b$ (please refer to Supplementary Fig. 24), two arrangements have no overlapping AVs. Accordingly, a big circular pore (red) was born automatically at the center of the octagonal arrangement under 190-V AV (Fig. 4b). The mixture arrangement is scalable, as evidenced by Fig. 4g where two spacing-different arrangements alternatively tile the whole plane. All experimental results are matched with the prediction, thus validating this hypothesis.

Within spacing-different mixture arrangements, pores with additional shapes were realizable and further tuned by adjusting

AV. For example, besides circular pores (green) in the basic arrangement, elliptical pores (cyan) were observed in the accessory arrangements (Fig. 4a, b). Smith et al. explained that the unique shape with higher structural asymmetry stems from the biaxial compressive stress competition between neighboring arrangements[21], as further evidenced by different anodization rates under nonuniform EF distribution therein (Supplementary Fig. 25). Likewise (see Supplementary Fig. 26), three-pointed star-shaped pores (blue, Fig. 4c) were realized in the centered tetragonal arrangement at 140-V AV. The accessory arrangements can be inserted in the form of line (Fig. 4d), area (Fig. 4e), and even complex patterns (Fig. 4f) as well, achieving other shapes such as isosceles triangular (yellow), rectangular (orange), and semi-circular (purple). Similarly, these new shapes are changeable by adjusting AV. For example, the pores owning the same spatial configuration (Supplementary Fig. 27) were varied from the non-bent (yellow, Fig. 4d) to the wall-externally-bent (yellow, Fig. 4f) triangular shape as increasing AV from 190 to 210 V. The successful mixing of spacing-different arrangements by appropriately choosing AVs will diversify the spatial configuration of AAO pores, thus giving rise to pores with designable shapes that are otherwise unobtainable in conventional spacing-identical configurations.

**Combining different sets of pores with independent shape designability.** In addition to well-defined structural controlling for one set of pores, we also generalized the designability of in-plane/out-of-plane pore shape and spacing-different spatial configuration into pore combinations. Figure 5b, d shows that by a wet-chemical etching step (Supplementary 28)[25], a 2nd-set

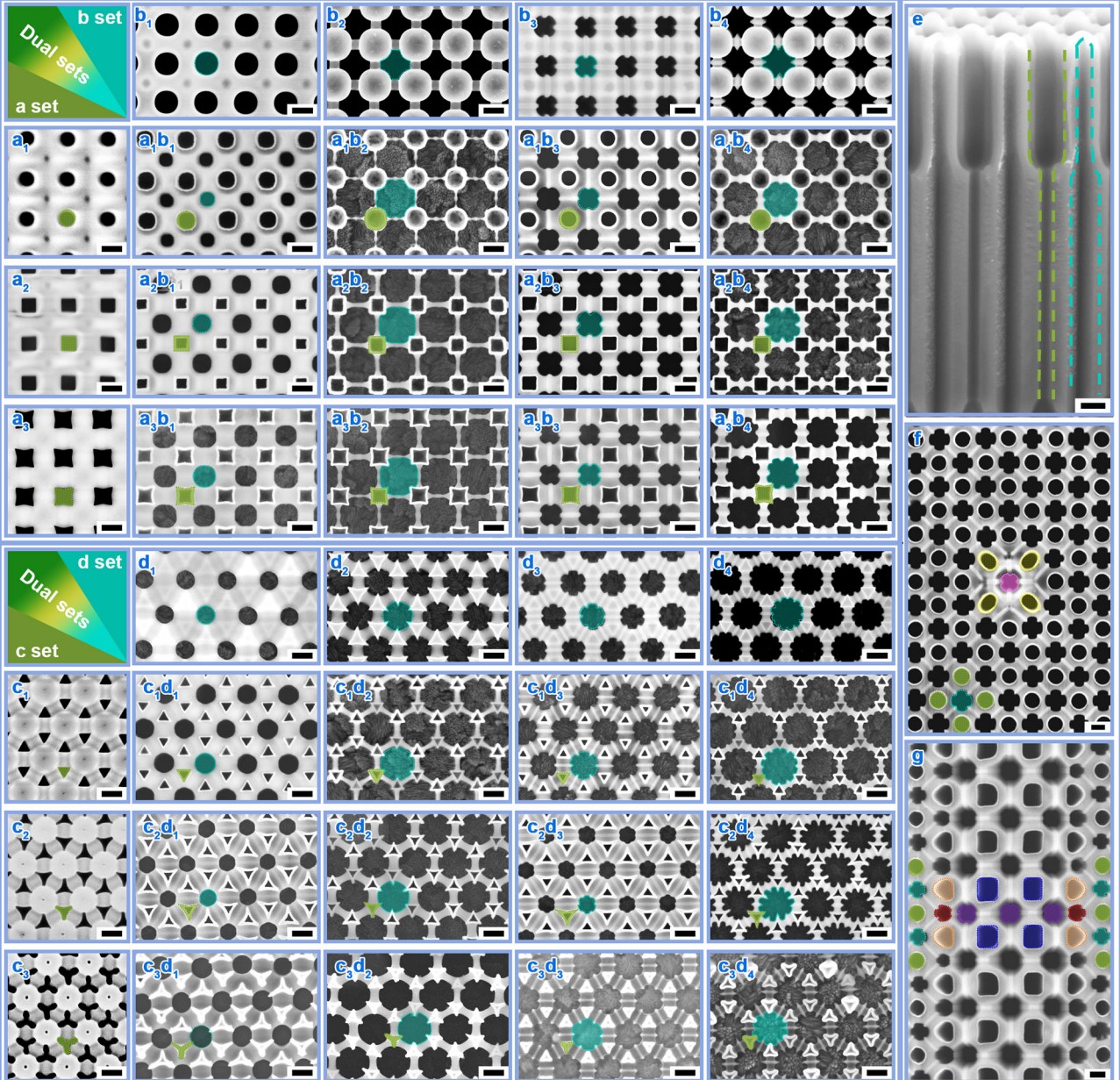

**Fig. 5 Templates with shape-designable pore combinations. $a_i$ $b_j$,** Pore combinations, with the 1st-set pores (of tetragonal arrangement, green color) in $a_i$ (similar to the pores in Fig. 2b–d) and the 2nd-set pores (of tetragonal arrangement, cyan colors) in $b_j$. The 1st-set and 2nd-set pores were formed before and after the wet-chemical etching procedure, respectively (please refer to Supplementary Fig. 28). The 2nd-set pores demonstrated four different shapes: when etching in NaOH solutions, the pores were isotropically enlarged from a circular shape (Fig. 5b$_1$) to a circular shape truncated with four voids (Fig. 5b$_2$); while using $H_3PO_4$ solutions, the pores were selectively etched from a 4four-edged (Fig. 5b$_3$) to an eight-edged cross shape (Fig. 5b$_4$). $c_m$ $d_n$, Pore combinations, with the 1st-set pores (of hexagonal arrangement, green color) in $c_m$ (similar to the pores in Fig. 2e–g) and the 2nd-set pores (of trigonal arrangements, cyan colors) in $d_n$. The 2nd-set pores also demonstrate four shapes: a circular shape (Fig. 5d$_1$) and a circular shape with 6 voids (Fig. 5d$_2$) etched in NaOH solutions, as well as a six-edged cross shape (Fig. 5d$_3$) and a 12-edged cross shape (Fig. 5d$_4$) etched in $H_3PO_4$ solutions. **e,** Pore combinations with two sets of out-of-plane dual-shape pores outlined by green (1st-set pore) and cyan (2nd-set pore) dashed lines. The anodization for growing the 1st-set pores was performed under sequential AVs of 140→200 V. **f, g** Multi-shape pore combinations within spacing-different mixture arrangements, stemming from the tetragonal templates mixed with point (Fig. 4a) and area (Fig. 4e) units of long-spacing arrangements. Colors of green/yellow/orange/blue and colors of cyan/pink/purple/red indicate pores formed before and after the wet-chemical etching procedure, respectively. Scale bars: 200 nm.

of pores can always be generated in a specific configuration and reshaped into four shapes regardless of different-AV induced variation in oxide. By assembling the anodized 1st-set pores (green) in Fig. 5a, c and the etched 2nd-set pores (cyan) in Fig. 5b, d into one matrix, we have achieved $2 \times 3 \times 4$ pore shape combinations, as summarized in Fig. 5a$_i$b$_j$ ($i = 1$–3, $j = 1$–4) and Fig. 5c$_m$d$_n$ ($m = 1$–3, $n = 1$–4). These pore combinations exhibit

large-area uniformity as well (Supplementary Figs. 29, 30). Besides the in-plane shape-designable pore combinations, we can achieve out-of-plane shape-designable pore combinations as well (Fig. 5e). The out-of-plane shape designability of the 2nd-set pores (outlined by cyan dashed line) arises from the fact that the pore walls formed at different AVs have inhomogeneous composition and thus etching rates (Supplementary Fig. 31)[22].

Taking this concept a step further, we realized different-set multi-shape pore combinations within spacing-different mixture arrangements (Fig. 5f, g). For example, starting from the mixture array with point of long-spacing arrangement (Fig. 4a), wet-chemical etching gave rise to two types of pores with the same four-edged cross shape but dissimilar orientations (cyan and pink), producing a two-set 4-shape template. Similarly, by etching the mixture array with area of long-spacing arrangements (Fig. 4b), pores of differently-oriented four-edged cross (cyan and purple) and three-edged cross (red) shapes were obtained, leading to two-set 6-shape templates together with the 1st-set pores of circular (green), rectangular (blue), and isosceles-triangular (orange) shapes (Fig. 5g).

In general, the total number of structural selection for templates with one-set pores and pore combinations could be estimated by the following equation:

$$f(x, y, z) = \sum_{i=1}^{x} (y_i + z_i + y_i \times z_i) \tag{1}$$

where $i$: the serial number of a specific spatial configuration of pores; $x$: the number of spatial configurations for the 1st-set pores which in principle need to satisfy the criterion that all constituent arrangements have an intersectional AV range; $y$: the number of the in-plane and out-of-plane shapes of the 1st-set pores determined by AV adjustable range and AV adjustable sequence; $z$: the number of the 2nd-set pore shapes controlled by wet-chemical etching. This equation provides a designing blueprint for AAO templates. Following this blueprint, AAO templates with broad selectivity and high controllability could be purposely diversified.

**Fabrication and application of well-defined nanostructures replicated from designable templates.** Template with designability of in-plane/out-of-plane pore shape, spatial configuration and pore combination should be the key for realizing well-defined nanostructures. To illuminate that various well-established material-synthesis techniques (e.g., atomic layer deposition, physical vapor deposition, and electrodeposition) are also applicable to designable templates[31–34], we fabricated 11 nanostructure arrays (please see Supplementary Figs. 32–39) with the assistance of three typical templates with 1st-set circular-shaped pores (Figs. 1d and 5a₁), 2nd-set four-edged cross-shaped pores (Fig. 5b₃), and the corresponding two-set pore combination (Fig. 5a₁b₃). Notably, designable templates are also compatible with other template-based synthetic techniques such as on-wire lithography and coaxial lithography[35–37], leading to many other well-defined nanostructures. The number of well-defined nanostructures stemming from designable templates can be described by:

$$f(x, y, z, a, b, c) = \sum_{i=1}^{x} (a_i \times y_i + b_i \times z_i + c_i \times (y_i \times z_i)) \tag{2}$$

where $a, b, c$: the numbers of the nanostructures replicated from the 1st-set pores, the 2nd-set pores, and the two-set combinations respectively (e.g., $a, b, c$ in Supplementary Fig. 32 are 3, 3, and 5); $i, x, y$, and $z$ have the same definitions as those in the Eq. (1). These well-defined nanostructures with high freedom of structural designability could result in some unique properties. Furthermore, external stimuli (e.g., capillary force, light, magnetic field, and heat) will further adjust these properties in a dynamic way[7]. Such promising features should be in favor of device utilization. Here we show three envisaged applications of nanostructures arrays prepared by the designable templates.

The first application is to optimize surface-enhanced Raman spectroscopy (SERS) using five hexagonal arrays of in-plane shape-different Ag nanoparticles (Fig. 6a and Supplementary Fig. 40). Figure 6d shows Raman spectra of Rhodamine 6G molecules chemisorbed on Ag nanoparticles. Compared to the nanoparticles with externally-bent shape (S1), the nanoparticles with non-bent triangular shape (S2) demonstrated a noticeable enhancement in Raman peak intensity. The highest intensity was achieved for the nanoparticles with internally-bent shape (S3), for example, the peak intensity at 1650 cm$^{-1}$ increased by approximately 6.6 times relative to that of S1. With further increasing of the internally-bent amplitude (S4 and S5), the SERS intensities were diminished. Finite-difference time-domain (FDTD) simulations demonstrate that hot spots with strong EFs, stemming from the plasmonic resonance effect[38], are situated around the vertexes of triangles (Fig. 6b). Particularly, the trend of the maximum EF values at the hot spots for five samples is consistent with that for the SERS intensity variation (Fig. 6c), implying that EFs enhanced by the designable nanoparticle shapes should be the determinative factor of SERS performance.

Then, we investigated the out-of-plane shape-designable Au nanowires as broadband light absorbers. We fabricated five samples (S6– S10) which included 1–5 shapes respectively, as schematically represented by the insets of Fig. 6h. A nanowire of S10 is shown in Fig. 6e, comprising five different shapes in the axial direction. An overall observation from the optical photographs is the gradual color variation from red to black (Fig. 6f), proving that the reflection of light impinging upon Au nanowires was suppressed with increasing shapes of nanowires. Light absorption efficiencies in the visible range increase monotonically from S6 to S10 (Fig. 6h), which agrees well with the visual observation. Figure 6g shows that strong EFs induced by the plasmonic resonance at short wavelengths (e.g., 500 and 540 nm) are situated at the top parts of the nanowires, that is, in the vicinity of the circular, superelliptic, and square segments. Regarding the long-wavelength illumination, the territory with EF enhancement moves down along the axis (e.g., at 580 and 750 nm) and finally centralizes around the star-like and cross-shaped segments (e.g., at 900 nm). Thus, multiple plasmonic modes are excited within different segments at various wavelengths and work complementarily to achieve strong light absorption in a broad regime.

Finally, we exploited the independent controlling over two-set TiO₂-nanotubes/Au-nanowires combination to promote photoelectrocatalysis. Circular (S11), square (S12), and star (S13) shapes were fabricated for optimizing TiO₂ nanotubes (Fig. 6i), which obtained different H₂ generation rates (Fig. 6k). The discrepancy is primarily ascribed to the variation in light trapping capability (Fig. 6j, l). Then, two shapes of four-void circular (S14) and eight-edged cross (S15) were fabricated for optimizing Au nanowires (Fig. 6i and Supplementary Fig. 41). Reliable H₂ growth was detected for the combinations under visible light, especially for S15 with 4.2 ± 0.7 µmol h$^{-1}$ cm$^{-2}$ (Fig. 6k). This long-wavelength response arises from plasmonic hot-electron injection from Au nanowires to TiO₂ nanotubes[39], as identified by three absorption peaks at 600, 680, and 740 nm (Fig. 6l). Figure 6j shows that EF enhancement tends to concentrate at the voids of Au nanowires, which accounts for the superior performance of S15 over that of S14 due to the occupation of twofold voids in Au nanowires. With optimization for both materials, S15 obtained a H₂ growth rate of 17.5 ± 1.5 µmol h$^{-1}$ cm$^{-2}$ under AM 1.5 G illumination, which resulted in about 3.5 times enhancement relative to S11.

## Discussion

This work advances the functionalities of AAO templates in controlling in-plane/out-of-plane shape, spatial configuration, and combination of pores. We developed a strategy to control the in-plane shapes of AAO pores by intentional design of unequal

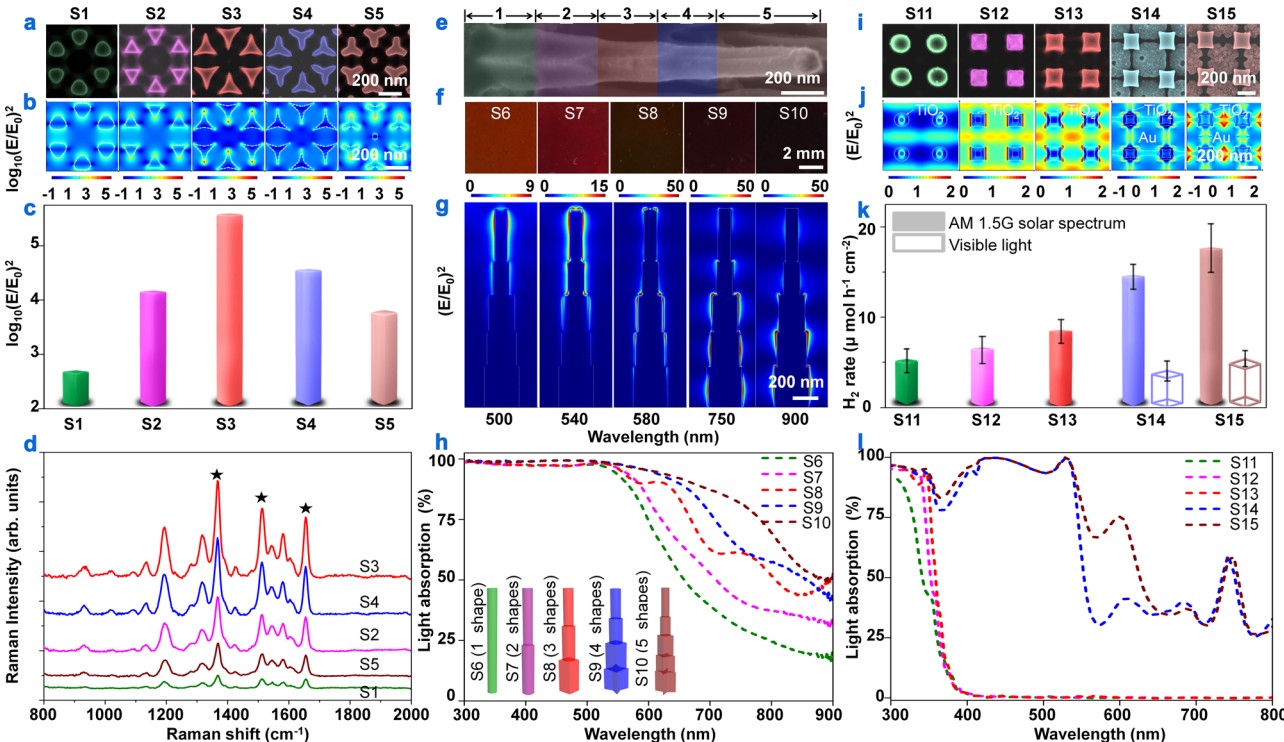

**Fig. 6 Applications of well-defined nanostructures.** SERS characterization with five hexagonal arrays of in-plane shape-different Ag nanoparticles (S1–S5). **a** SEM images of unit cells and **b** FDTD-simulated EF intensities (logarithmic scale, 532-nm wavelength illumination). **c** The maximal magnitudes of EF intensities in **b**. **d** SERS spectra with three main peaks at 1363, 1508, and 1650 cm$^{-1}$ for Rhodamine 6G molecules chemisorbed on Ag nanoparticles. Light absorbers using five tetragonal arrays of out-of-plane shape-different Au nanowires (S6–S10). **e** SEM image of 5-shape-combined Au nanowire of S10 combining five shapes in the axial direction (from left to right): cross, star, square, superelliptic, and circular. **f** Optical photographs of S6–S10, involving Au nanowires of 1–5 shapes. **g** FDTD-simulated EF intensities for S10 under different-wavelength illumination. **h** Measured light absorption spectra. Insets: schematic illustration for nanowires of S6 to S10, featuring circular, circular + superelliptic, circular + superelliptic + square, circular + superelliptic + square + star, and circular + superelliptic + square + star + cross shapes. **i–l** Photoelectrocatalytic H$_2$ production using shape-different TiO$_2$ nanotubes and Au nanowires (S11–S15). **i** SEM images (left to right): TiO$_2$ circular nanotubes, TiO$_2$ square nanotubes, TiO$_2$ star nanotubes, TiO$_2$ star nanotubes/Au four-void-circular nanowires, TiO$_2$ star nanotubes/Au eight-edged-cross nanowires. **j** FDTD-simulated EF intensities for TiO$_2$ nanotubes (linear scale, 350-nm-wavelength illumination) and TiO$_2$-nanotubes/Au-nanowires (logarithmic scale, 600-nm-wavelength illumination). **k** H$_2$ production rates illuminated by AM 1.5 G spectrum and visible light. Error bars show standard deviations. **l** Calculated light absorption spectra in TiO$_2$ nanotubes and TiO$_2$-nanotubes/Au-nanowires.

anodization rates at different AVs and firstly realized pores with designable in-plane shapes by easily adjusting AV in specific ranges, thus reducing the high cost of mask/stamp fabrication in conventional lithography for fabricating shape-different nanostructures especially with small feature sizes (e.g., <400 nm). And the fabricated out-of-plane shape-designable pores (and resultant nanostructures) arrays that combine different in-plane shapes along the axial are not easily obtainable by using other techniques. Unlike conventional AAO template in which all pores feature identical spacings in order to satisfy the linear spacing-AV relation[21], we found that pores of different spacings can be mixed into one matrix as long as the constituent pore arrangements have an intersectional AV range, highly enriching spatial configuration (and resultant shapes) of pores. The designability in in-plane/out-of-plane shape and spacing-different arrangements is applicable to each set of pores in different-set templates, leading to a large series of designable pore combinations. We believe that the broad selectivity and high controllability of AAO templates will pave the way towards well-defined nanostructuring over a library of nanostructure arrays, which may be attractive to a wide spectrum of technological applications.

## Methods

**Atomic layer deposition.** Atomic layer deposition (ALD) was performed with a Picosun SUNALE™ R150 ALD system. For TiO$_2$ growth, the ALD deposition was carried out at 300 °C with a growth rate of about 0.05 nm per cycle. A typical growth cycle includes 0.1 s pulsing of titanium (IV) chlorides, 5 s purging of nitrogen, 0.1 s pulsing of deionized (DI) water, and 5 s purging of nitrogen. For SnO$_2$ growth, the ALD deposition was carried out at 250 °C with a growth rate of about 0.03 nm per cycle. A typical growth cycle involves 0.5 s pulsing of tin (IV) chloride, 10 s purging of nitrogen, 0.5 s pulsing of DI water, and 10 s purging of nitrogen. For metallic Pt, the ALD deposition was carried out at 300 °C with a typical growth cycle of 1.5 s pulsing of Pt(MeCp)Me$_3$, 30 s filling of nitrogen, 20 s purging of nitrogen, 1.5 s pulsing of oxygen, 30 s filling of nitrogen, and 20 s purging of nitrogen.

**Electrochemical deposition.** Two-electrode configuration was exploited for all electrochemical deposition using an electrochemical workstation (BioLogic). Ni deposition was performed in an aqueous solution (0.12 M NiCl$_2$, 0.12 M Ni$_2$SO$_4$, and 0.5 M H$_3$BO$_3$) at a constant current density of 2 mA/cm$^2$ with a Ni plate as counter electrode. Au and Ag deposition were conducted at a constant current density of 1 mA/cm$^2$ with counter electrode of a Pt foil in the plating solutions.

**Material etching.** The unanodized aluminum foil was etched in a mixture solution including 3.4 g CuCl$_2$, 100 ml HCl, and 100 ml DI water at room temperatures. The anodic aluminum oxide (AAO) template was wet-chemically etched in 0.5 M H$_3$PO$_4$ solutions or 0.1 M NaOH solutions. For keeping a constant etching rate for AAO template, the wet-chemical etching was always conducted at a constant temperature of 30 °C. Dry etching upon AAO template was performed by an argon ion milling system (Gatan PECS™) with an etching power of 4.5 kV, rotation frequency of 2 Hz, and tilted angle of 60°.

**Fabrication of Ni imprint stamp with clover-like nanopillars.** To obtain a Ni stamp with a tetragonal array of four-leaf clover-like nanopillars, a commercially

available Si mold patterned with circular nanoconcaves on its surface was exploited, which are of 400 nm spacing and tetragonally arranged. First, the Si mold was cleaned in a mixture solution ($H_2O_2$:$H_2SO_4$, 1:3) at 100 °C for 1 h. After cleaning in DI water, the Si mold was treated using 3-aminopropyltriethoxysilane (1.0 v% in $CH_3CH_2OH$) at 60 °C for 1.5 h. Then, a gold layer (thickness of about 15–25 nm) was evaporated onto the nanopatterned surface of the Si mold by physical vapor deposition (PVD) system, serving as a conductive layer in the following electro-chemical deposition. Afterwards, a thick Ni layer was electrodeposited at a constant current density of 2 mA/cm$^2$. After the electrochemical deposition, the Ni foil was dried by airflow and peeled off from the Si mold. Accordingly, a Ni foil equipped with a tetragonal array of circular nanopillars was obtained.

The as-obtained circular nanopillars were then utilized for patterning an aluminum foil surface by mechanical impressing under a pressure of 10 kN cm$^{-2}$ (Supplementary Fig. 5a$_1$). The imprinted area was then anodized in 0.4 M $H_3PO_4$ solutions at an AV of 160 V at 10 °C for 10 min, leading to a porous AAO template (Supplementary Fig. 5a$_2$). Afterward, polymethyl methacrylate (PMMA) was dripped onto the anodized area and dried naturally to form a supporting layer. After wet-chemically removing the unanodized aluminum foil (Supplementary Fig. 5a$_3$), the exposed AAO template was selectively etched in 0.5 M $H_3PO_4$ solutions for 90 min, and then covered by a gold layer (thickness of about 15–20 nm) by PVD (Supplementary Fig. 5a$_4$). With this conductive layer as working electrode, Ni electrodeposition was performed to form a thick layer (Supplementary Fig. 5a$_5$). After dissolving the supporting PMMA layer in dimethyl sulfoxide solution at 80 °C for 2 h and completely removing the AAO template in the $H_3PO_4$ solution for 5 h, a Ni stamp with a tetragonal array of four-leaf clover-like nanopillars was obtained (Supplementary Fig. 5a$_6$). The as-fabricated nanopillars are shown in Supplementary Fig. 5b, c.

Similarly, we fabricated another Ni imprint stamp with a hexagonal array of three-leaf clover-like nanopillars (Supplementary Fig. 11). The fabrication process (Supplementary Fig. 11a) was nearly the same as that for the Ni stamp with a tetragonal array of four-leaf clover-like nanopillars, except for that it started with a Si mold patterned with a trigonal array of circular nanoconcaves. Finally, a Ni stamp with a hexagonal array of three-leaf clover-like nanopillars was constructed, as shown in Supplementary Fig. 11b, c.

**Anodic anodization of aluminum**. For predetermining nanoconcaves on aluminum foils to guide pore evolution during anodization, Ni stamps with arrayed nanopillars were exploited for engineering electro-polished aluminum foils by an imprinting system at a constant pressure of 10 kN cm$^{-2}$ for 3 min. Aluminum foils patterned with nanoconcaves of hexagonal arrangement and $400/\sqrt{3}$ nm spacing were always anodized in 0.4 M $H_3PO_4$ solutions regardless of the applied AV. Regarding aluminum foils patterned with nanoconcaves of tetragonal arrangement and 400 nm spacing, the anodization electrolytes were selected on the base of AVs. That is because higher AVs accelerate Al anodization (i.e., high anodization current) and results in accumulation of heat (and temperature increasing), which causes electrolytic breakdown of AAO template and prevents the formation of pores with high aspect ratios. Therefore, as AVs were lower than 180 V, anodization was conducted in 0.4 M $H_3PO_4$ solutions; while beyond 180 V, a mixture solution including 3 ml $H_3PO_4$, 300 ml ethylene glycol, and 600 ml DI water was exploited as anodization electrolyte. The presence of ethylene glycol and the reduction of $H_3PO_4$ concentration can effectively mitigate electrolytic breakdown of AAO template at high AVs.

**Synthesis of nanostructure arrays with AAO templates**. To construct Ag nanoparticles with in-plane different shapes (S1–S5 in Fig. 6a–d), the lab-made Ni stamp with a hexagonal array of three-leaf clover-like nanopillars was exploited to engineer aluminum surface. At different AVs (e.g., 150, 140, 120, 100, and 80 V), the nanoconcaves inheriting the geometrical features of Ni three-leaf clover-like nanopillars were evolved into pores of different shapes during anodization. After anodization, a 50-nm-thick Ag layer was evaporated onto the anodized area by PVD, serving as a conductive layer. Then, Ag electrodeposition was performed to form a thick substrate. After wet-chemically removing the unanodized aluminum foil and the AAO template, arrays of in-plane shape-different Ag nanoparticles were obtained.

For fabricating out-of-plane shape-different Au nanowires (S6 to S10 in Fig. 6e–h), the Ni stamp with tetragonally arranged four-leaf clover-like nanopillars was exploited to pattern aluminum foils. Upon the imprinted area, sequential anodization was conducted at different AVs to tune the out-of-plane pore shape. Taking Au nanowires combining five shapes (S10) as an example, five-step anodization at 120 V for 20 min → 140 V for 7.5 min → 160 V for 5 min → 180 V for 15 min → 200 V for 10 min was sequentially performed to fabricate five-shape combined pores. Then, a 5-nm-thick $TiO_2$ layer was deposited along the pore walls by ALD to tune the dielectric environment of the following Au nanowires and meanwhile endow the high-aspect-ratio Au nanowires with mechanical robustness. To form a supporting substrate for Au nanowires, we deposited a metallic layer (5-nm-thick Ti and 20-nm-thick Au) onto the template, serving as working electrode in the following Au and Ni electrodeposition. After completing Au and Ni electrodeposition, the unanodized aluminum foil on the backside was wet-chemically removed, followed by ion-milling off the pore barrier. When the Au plating solution can penetrate through pores and touch the conductive substrate,

Au electrodeposition was carried out, and the resultant Au nanowires replicated the structural features of pores. Finally, an array of Au nanowires combining five shapes in the axial direction was obtained after wet-chemically dissolving the AAO template in NaOH solutions. Using the same way, we fabricated pores combining one, two, three, and four shapes as well as the resulting Au nanowire arrays. Taking changeable pore-elongation rates at different AVs into consideration, the anodization parameters for the other four samples were 200 V for 50 min (S6), 180 V for 38 min → 200 V for 25 min (S7), 160 V for 8 min → 180 V for 25 min → 200 V for 17 min (S8), and 140 V for 9 min → 160 V for 6 min → 180 V for 19 min → 200 V for 12.5 min (S9) to maintain the same length for all Au nanowires.

For constructing $TiO_2$ nanotubes and $TiO_2$-nanotubes/Au-nanowires combinations (S11–S15 in Fig. 6i–l), we exploited the Ni stamp with a tetragonal array of four-leaf clover-like nanopillars to engineer aluminum foils. Under the AVs of 140, 160, and 200 V, the 1st-set pores evolved into star, square, and circular shapes, respectively. A 40-nm-thick $TiO_2$ layer was then grown along the pore walls by ALD to serve as semiconductor photocatalyst. After ion-milling off the $TiO_2$ layer on the topmost surface, a metallic layer (5-nm-thick Ti and 20-nm-thick Au) was evaporated onto the top surface by PVD, followed by Ni electrodeposition to form a supporting substrate. Afterwards, the unanodized aluminum foil and the AAO template were wet-chemically etched, leading to $TiO_2$ nanotubes with different shapes (S11–S13). For obtaining $TiO_2$-nanotubes/Au-nanowires arrays, the 2nd-set pores were opened at the fourfold junction sites of neighboring 1st-set pores using NaOH solutions for 20 min. The opened 2nd-set pores were then isotropically etched in NaOH solutions to form a circular shape truncated with four voids or selectively etched in $H_3PO_4$ solutions to form an eight-edged cross shape. With the exposed conductive substrate as working electrode, Au electrodeposition was conducted in the 2nd-set pores to form shape-different Au nanowires, which interfaced with the $TiO_2$ nanotubes in the 1st-set pores. Finally, $TiO_2$-nanotubes/Au-nanowires arrays were obtained after dissolving the AAO template in NaOH solutions (S14 and S15).

**SEM and FIB cutting**. All SEM analysis of structural morphology and nanoscale cutting in this work were performed by Focused Ion Beam Scanning Electron Microscopes (ZEISS).

**SERS measurement**. The preparation for surface-enhanced Raman spectroscopy (SERS) measurement was performed by immersing arrays of in-plane shape-different Ag nanoparticles into Rhodamine 6 G aqueous solutions with a concentration of 10$^{-6}$ M. The immersion procedure was carried out for 4 h to assure that sufficient R6G molecules can be chemisorbed on the surface of Ag nanoparticles, especially the hot spots under electromagnetic excitation (namely, the vertices of triangular nanoparticles). Before measurement, all samples were dipped into DI water for 30 s and dried with nitrogen flow. SERS signals spectra with three main peaks at 1363, 1508, and 1650 cm$^{-1}$ were collected by the NTEGRA Spectra system (NTMDT) equipped with a laser source of 532-nm wavelength. The measured data were treated with baseline correction to extract SERS spectra from the broad background.

**Optical measurement**. UV–vis–NIR spectrometer (Varian Cary 5000) was exploited to characterize light trapping performance. All optical measurements were performed with an illumination spot size of ~20 mm$^2$ at normal incidence without polarization. Considering opaqueness for all samples, the experimental light absorption spectra in this work were calculated by 100% − R% where R% represents the light reflection efficiency.

**Photoelectrocatalytic measurement**. The hydrogen production was measured by a gas chromatography analyzer (Agilent Micro-GC). For facilitating redox reaction, Pt nanoparticles were formed at the surface of $TiO_2$ nanotubes by ALD. The photocatalysts of $TiO_2$ nanotubes and Au-nanowires/$TiO_2$-nanotubes combinations were immersed into a mixture solution of 1:4 v/v methanol/DI water. Argon-flow purging was kept for 1 h to eliminate the dissolved oxygen from the aqueous solution. During measurement, the photocatalytic samples were illuminated by simulated AM 1.5 G solar spectrum or visible light with a 420-nm long-pass filter. The hydrogen production rates for all samples were normalized to the corresponding active areas.

**COMSOL simulation**. To characterize the electric field distribution in aluminum foils at the initial stage of anodization, we exploited the ACDC module (simple resistor) from COMSOL Multiphysics (5.1 version) for simulation. The geometrical features of surface-patterned aluminum foils exploited in simulation were set according to the experimental observation. Quadruple and sextuple unit cells with periodic boundary conditions were exploited for calculation, corresponding to the experimental nanoconcaves of tetragonal and hexagonal arrangements, respectively. A voltage difference was applied between the top surface and bottom of the aluminum foil to simulate the external AV.

**FDTD simulation**. Optical simulations were performed with a commercial software package (FDTD solutions, Lumerical Solutions). Three-dimensional layouts

were exploited for all numerical calculations. The plane-wave light source propagating in the −z direction was used to illuminate nanostructure arrays, which were situated on a substrate parallel with the xy plane. Periodic boundary condition was set along the x and y directions, and perfectly matched layers were used to cut the z axis. The geometries of nanostructure arrays in simulation were measured from the corresponding SEM images. The electric field maps were recorded by two-dimensional frequency-domain field profile monitors. For calibrating the light absorption efficiencies (A%), two two-dimensional power monitors were exploited: one was placed at the bottom of the nanostructure array to measure the transmitted power (T%), and the other was located above the light source to measure the reflected power (R%). The light absorption spectra were obtained by 100% − T% − R%. Optical properties of Ag and Au in simulation were obtained from Johnson and Christy[40] and optical data reported in Siefke's paper were used for modeling $TiO_2$[41].

## Data availability

All of the data supporting the conclusions are available within the article and the Supplementary Information. Additional data are available from the corresponding authors upon reasonable request.

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

## Acknowledgements

The authors are grateful to H. Romanus, Dominik Flock, and Diana Herz for help with FIB cutting, EDX analysis, and argon ion milling. The authors thank J. Döll and A. Konkin for their support in optical characterization. The authors thank H. Zhao, H. Zhang, and M. Sommerfeld for technical discussion. This work is financially supported by the German Research Foundation (DFG: LE 2249/5-1) and Sino-German Center for Research Promotion (GZ 1579).

## Author contributions

R.X. and Y.L. conceived the concept. Y.L. supervised the project. R.X. conducted the experiments and simulations. Z.Z. helped with device measurement. All authors participated in the discussion on the data. R.X. and Y.L. wrote the manuscript.

## Funding

## Competing interests

The authors declare no competing interests.
