## [Peer Review File · Nature Communications]

Title: Well-defined nanostructuring with designable anodic aluminium oxide templateREVIEWER COMMENTS

Reviewer #1 (Remarks to the Author):

This is a very nice paper in which strategies for the three-dimensional tailoring of AAO membranes are comprehensively devised. Moreover, interesting applications of replicas of the three-dimensional AAO structures are reported. It is also apparent that the authors put a lot of efforts in this work. Therefore, I strongly recommend the publication of this work.

There are some editorial points that could be improved. What is an „uneven anodization rate“? What is meant by “internally” and “externally bent walls”? Are concave and convex walls meant?

At first, it remains unclear and unmentioned that hard imprint lithography of the Al substrate to be anodized is the apparent starting point and that the topography of the surface of the stamps is a crucial feature for the following structure formation. This very crucial step needs to be highlighted in the introduction – otherwise, it is hard to follow the manuscript. The impact of the design of the imprint stamps and the role of the imprint step should be addressed. The naming of the topographic features generated in the Al substrate as “nanodents” is somewhat misleading. The scheme integrated in Figure 1 should be a Figure on its own.

Also, it should be mentioned, which electrolyte was used and which role the electrolyte plays. In line 96, all of a sudden PO₄ anions appear. Where do they come from?

Reviewer #2 (Remarks to the Author):

The authors presented an interesting investigation of dynamic anodization voltage control induced complexity-diversified alumina pattern formation on the pre-pressed aluminum foil. The as-demonstrated work successfully overcomes the limited access of the current techniques in literature to in-plane and out-of-plane shape variation/control, realizing a remarkable level of complexity control over the AAO template fabrication process, which is indeed a breakthrough in the template based nanostructuring for well-defined nanofeature arrays of 3D heterogeneity, complexity, as well as periodicity. Overall, the work is soundly presented with solid data analyses and elaborations with clearly articulated methodologies and reasoning, combining both simulation and experimental approaches. It is indeed worthwhile publishing given the demonstrated significant progress built on the authors' prior work in Nature Nanotech. a few years ago. A few minor comments here: 1) Some more discussion over the overall strategies of the voltage control with respect to different pattern selection/determination could be helpful to the generalization of the new techniques besides the normal/current anodization approaches. 2) The electrical field simulation is well matched with experimental results, any experimental deviations that could be introduced due to various experimental factors? Or any examples of failures that could help shed some lights on the general practices? Some discussions along the line could be helpful to the adoption of the new technique here. 3) The demonstration of the out of plane shape variation through dynamic voltage control is well done, but how to fully utilize it as a template seems to be less elaborated, given the demonstrations are quite limited to the photon-responsive

examples. Any other potentially carriers?

Reviewer #3 (Remarks to the Author):

Yong Lei and his co-workers report a method to prepare complex yet high-fidelity nanoporous AAO templates by heterogeneous aluminium anodization rates with varied anodization voltages. Authors clearly introduced previous challenges and motivation for this work and demonstrated experimental evidence for their claim. As cited in ref. [22], authors previously reported a large-area nanofabrication of binary pored AAO templates in 2017. While there have been a number of applications of nanostructured AAO templates, reports on new strategy for nanofabrication has been limited. Hence, I enjoyed the reading this manuscript. In particular, I am impressed by the mix-and-match of different cross-sectional geometry for nanopores along the z-direction as well as spatial arrangements in x-y plane. Obviously, these new fabrication capabilities will contribute to overcome previous limitations and may open new horizons in various fields. Also, authors already performed extensive parameter studies and the experimental results are clearly supporting the main claim. Hence, I am favorable for publication of this manuscript in Nature Communications.

However, considering free-form 3D nanoarchitectures produced by Nanoscribe, I think that “all-in-one platform” in the Abstract is overselling and ask authors to tone-down.

While it's for microstructures and based-on a growth mechanism, below reference is relevant and worthwhile to cite in the Introduction as an effort to prepare uneven cross-sectional shapes for small-scale fabrication.

<https://www.nature.com/articles/ncomms7584>

Stimuli-responsive micro/nanostructures are based-on non-uniform responses upon external stimuli. Thus, this work will have a great potential in that field. Below reference is worthwhile to cite in the Introduction and mention potential applications in stimuli-responsive systems.

<https://onlinelibrary.wiley.com/doi/epdf/10.1002/pol.20210311>

Point-to-point responses to reviewers' comments & description of the change we have made to the manuscript to address these comments

Re: Manuscript ID NCOMMS-21-32423-T

We highly appreciate the detailed and constructive comments put forth by the reviewers, and have revised the manuscript accordingly, with all the reviewers' concerns being addressed. This letter details our point-by-point responses to the comments. The revised texts are written in blue font in our revised manuscript and supporting information.

To Reviewer 1:

Comments:

This is a very nice paper in which strategies for the three-dimensional tailoring of AAO membranes are comprehensively devised. Moreover, interesting applications of replicas of the three-dimensional AAO structures are reported. It is also apparent that the authors put a lot of efforts in this work. Therefore, I strongly recommend the publication of this work.

Answer: We highly appreciate the reviewer for the comments.

1. *There are some editorial points that could be improved. What is an „uneven anodization rate”? What is meant by “internally” and “externally bent walls”? Are concave and convex walls meant?*

Answer: Many thanks for the comments.

1) “Uneven anodization rate” has been changed to “unequal anodization rate”.

2) For better understanding, “internally” and “externally bent walls” are schematically illustrated in Supplementary Fig. 1 on page 2 of the revised Supplementary Information.

Supplementary Fig. 1 | Schematic illustration of in-plane shape-different pores. (a) Square and (b) triangle with (from left to right) internally-bent walls, non-bent walls and externally-bent walls.

The revised statement about “internally” and “externally bent walls” on pages 3-4 of the revised manuscript:

“the in-plane pore shape of the designable template can be continuously altered from polygons (*e.g.*, triangle and square) with internally-bent (*i.e.*, concave) walls to polygons with non-bent walls (*i.e.*, straight) and then to polygons with externally-bent (*i.e.*, convex) walls (please see Supplementary Fig. 1 for the schematic illustration of shape-different pores)

2. *At first, it remains unclear and unmentioned that hard imprint lithography of the Al substrate to be anodized is the apparent starting point and that the topography of the surface of the stamps is a crucial feature for the following structure formation. This very crucial step needs to be highlighted in the introduction – otherwise, it is hard to follow the manuscript.*

Answer: Thanks for the comment. According to your suggestion, the “hard imprint lithography” and “topography of the surface of the stamps” have been highlighted in the introduction on page 3 of the revised manuscript:

“Different from self-ordered two-step anodic anodization which only generates circular-shaped pores with close-packed honeycomb (*i.e.*, trigonal) arrangement, artificially nanoengineering Al-foil surface by hard imprint lithography can guide anodic anodization to form pores with desired structural parameters (*e.g.*, arrangement and interpore spacing) that highly depend on the surface topography of imprinting stamps”

3. *The impact of the design of the imprint stamps and the role of the imprint step should be addressed.*

Answer: We sincerely thank you for the suggestion.

The role of the imprint step is explained on page 4 of the revised manuscript:

“Given that nanoimprinting Al-foil surface with appropriate texture could guide the initiation of pores¹⁹, we hope to generate unequal aluminium anodization rates by introducing uneven-profiled four-leaf clover-like nanoconcaves onto surface (see layout of COMSOL simulation in Supplementary Fig. 2)”

The impact of the design of the imprint stamps is explained on page 6 of the revised manuscript:

“To discern the crucial role of inhomogeneous radial anodization rates (*i.e.*, uneven EF distribution) in shape designability, we fabricated circular nanoconcaves for reference by using a Ni circular-pillar stamp (Supplementary Fig. 8) and achieved homogeneous radial anodization rate (as evidenced by even EF distribution) on the side-walls of nanoconcaves (Supplementary Fig. 9d). Not surprisingly, the pore shape cannot be altered no matter what AV was applied (Supplementary Fig. 10).”

4. The naming of the topographic features generated in the Al substrate as “nanodents” is somewhat misleading.

Answer: Thank you for the comment. The naming of the topographic features generated in the Al substrate has been changed to nanoconcaves.

5. The scheme integrated in Figure 1 should be a Figure on its own.

Answer: According to your nice suggestion, the scheme integrated in Figure 1 has been a Figure on its own, as shown on page 27 of the revised manuscript.

6. Also, it should be mentioned, which electrolyte was used and which role the electrolyte plays. In line 96, all of a sudden PO4 anions appear. Where do they come from?

Answer: Many thanks for the suggestion. The used electrolyte and the role of the electrolyte plays are introduced on page 5 of the revised manuscript:

“Here, an arbitrary spacing of 400 nm is set for the preset nanoconcave array. Following the linear spacing-AV relation, the appropriate AV should be 160 V and accordingly aqueous solution of 0.4 M H₃PO₄ is used as anodization electrolyte to form nanoporous structures²⁴.”

To Reviewer 2:

Comments:

The authors presented an interesting investigation of dynamic anodization voltage control induced complexity-diversified alumina pattern formation on the pre-pressed aluminum foil. The as-demonstrated work successfully overcomes the limited access of the current techniques in literature to in-plane and out-of-plane shape variation/control, realizing a remarkable level of complexity control over the AAO template fabrication process, which is indeed a breakthrough in the template based nanostructuring for well-defined nanofeature arrays of 3D heterogeneity, complexity, as well as periodicity. Overall, the work is soundly presented with solid data analyses and elaborations with clearly articulated methodologies and reasoning, combining both simulation and experimental approaches. It is indeed worthwhile publishing given the demonstrated significant progress built on the authors' prior work in Nature Nanotech. a few years ago.

Answer: Thank you very much for the comments.

A few minor comments here:

1) Some more discussion over the overall strategies of the voltage control with respect to different pattern selection/determination could be helpful to the generalization of the new techniques besides the normal/current anodization approaches.

Answer: Thanks a lot for the comment. In the revised manuscript, we have further discussed (1) why we use constant-voltage anodization rather than constant-current anodization, (2) the appropriate AV range for pore-shape tuning regarding a specific array, (3) the appropriate AV range for a mixture arrangement.

The reason that we use constant-voltage anodization rather than constant-current anodization is introduced on page 4 of the revised manuscript:

“Here, we select potentiostatic anodization for structural controlling of pores in consideration of the linear relationship between AV and pore parameters (*e.g.*, interpore distance and pore diameter)¹⁷”

The appropriate AV range for pore-shape tuning regarding a specific array is described on page 7 of the revised manuscript:

“It is found that concerning a specific arrangement, the adjustable AV for designing pore shape is limited in an AV range (denoted as appropriate AV range), out of which (*i.e.*, in too-low AV and too-high AV ranges) the arrangement predetermined by nanoconcaves is broken (Supplementary Fig. 15). The three AV ranges are separated by two threshold values (Supplementary Fig. 16), which are empirically observed to

be V_a and $\sqrt{3}V_a$, where $V_a = 0.4 \text{ (V nm}^{-1}) \times L_h \text{ (nm)}$ and L_h is the interpore spacing of the hexagonal array (*i.e.*, $400/\sqrt{3}$ nm). In other words, AV thresholds can be derived from the linear spacing-AV relation, *i.e.*, $\text{AV (V)} = \text{spacing (nm)} \times 0.4 \text{ (V nm}^{-1})^{24}$, regarding two spacings (*e.g.*, L_h and $\sqrt{3}L_h$) of emerging arrays”

The appropriate AV range for a mixture arrangement (*i.e.*, spacing-different mixture arrangement, mixture of tetragonal and octagonal arrangements) is discussed on page 9 of the revised manuscript:

“we hypothesized that AVs of a spacing-different mixture arrangement should lie in an intersection of several appropriate AV ranges to simultaneously prevent the occurrence of new pores and the disappearance of existing pores for every constituent arrangement. To test this hypothesis, we mixed the above tetragonal arrangement with spacing of L_t (*i.e.*, 400 nm) with a larger-spacing ($\sqrt{2}L_t$) tetragonal arrangement (Supplementary Fig. 22, a to c). In theory, the appropriate AV ranges for two arrangements are ($V_b/\sqrt{2}$, $\sqrt{2}V_b$) and (V_b , $2V_b$) (see Supplementary Fig. 22, d to f for details), where $V_b = 0.4 \text{ (V nm}^{-1}) \times L_t \text{ (nm)}$.²⁴ The intersectional AVs locate from V_b (160 V) to $\sqrt{2}V_b$ (~226 V).”

“Regarding the octagonal arrangement with an appropriate AV range from $\sqrt{2.5}V_b$ to $\sqrt{5}V_b$ (please refer to Supplementary Fig. 24), two arrangements have no overlapping AVs”

2) The electrical field simulation is well matched with experimental results, any experimental deviations that could be introduced due to various experimental factors? Or any examples of failures that could help shed some lights on the general practices? Some discussions along the line could be helpful to the adoption of the new technique here.

Answer: We sincerely thank for this valuable comment. As well known, beside the anodization voltage, the pore features of anodized AAO templates are highly dependent on the preset texture of the Al-foil surface and the anodization electrolyte [DOI: 10.1021/cr500002z].

Firstly, we decorated the Al-foil surface with circular nanoconcaves to investigate the role of the preset shape of nanoconcanves. We found that the anodized pores had circular shape and were not alterable with the applied AV (Supplementary Figure 10). We then carried out electric field simulation to shed light on the underlying mechanism. In Supplementary Figure 9, the circular-shaped nanoconcaves demonstrate homogeneous radial anodization rate. Based on these results, we conclude that the uneven profile of the four-leaf clover-like nanoconcaves that cause inhomogeneous radial anodization rates plays a critical role in shape designability.

The importance of preset shape of nanoconcaves on Al-foil surfaces is described on page 6 of the revised manuscript:

“To discern the crucial role of inhomogeneous radial anodization rates (*i.e.*, uneven EF distribution) in shape designability, we fabricated circular nanoconcaves for reference by using a Ni circular-pillar stamp (Supplementary Fig. 8) and achieved homogeneous radial anodization rate (as evidenced by even EF distribution) on the side-walls of nanoconcaves (Supplementary Fig. 9d). Not surprisingly, the pore shape cannot be altered no matter what AV was applied (Supplementary Fig. 10)”

As for the anodization electrolyte, we found that as the AV was much higher than the value following the linear space-AV relation (DOI:10.1021/nl025537k), the AAO templates tended to suffer from electrolytic breakdown. To prevent its occurrence, we introduced organic solvent to slow down anodic anodization and meanwhile decreased the H₃PO₄ concentration of anodization electrolyte. The importance of anodization electrolyte is introduced on page 18 of the revised manuscript:

“Regarding aluminium foils patterned with nanoconcaves of tetragonal arrangement and 400 nm spacing, the anodization electrolytes were selected on the base of AVs. That is because higher AVs accelerate Al anodization (*i.e.*, high anodization current) and result in accumulation of heat (and temperature increasing), which causes electrolytic breakdown of AAO templates and prevents the formation of pores with high aspect ratios. Therefore, as AVs were lower than 180 V, anodization was conducted in 0.4 M H₃PO₄ solutions; while beyond 180 V, a mixture solution including 3 mL H₃PO₄, 300 ml ethylene glycol, and , and 600 ml DI water was exploited as anodization electrolyte. The presence of ethylene glycol and the reduction of H₃PO₄ concentration can effectively mitigate electrolytic breakdown of AAO template at high AVs.”

3) *The demonstration of the out of plane shape variation through dynamic voltage control is well done, but how to fully utilize it as a template seems to be less elaborated, given the demonstrations are quite limited to the photon-responsive examples. Any other potentially carriers?*

Answer: Many thanks for the comment.

As indicated in the introduction of the revised manuscript “nanostructures are subject to physical and chemical property variation as a function of their geometry and composition”, the out-of-plane shape-designable structures are capable of combining different physical and chemical properties that inherit from nanostructures with different geometrical features along the axial direction. The combination of different properties into one sample could be in favor of synthesis of materials with new properties such as magnetic

materials (DOI: 10.1039/C4CE00048J) and shape-memory alloys (DOI: 10.1038/nano.2017.91), as well as enabling applications with enhanced performance and/or broad operational conditions, for example, electrochemical sensor (DOI: 10.1016/j.talanta.2008.09.042), multi-modal nanolasing (DOI: 10.1038/nano.2017.126), biomedical applications (DOI: 10.1039/C9CS00011A), energy-related devices (DOI: 10.1039/C1PC90010B), and semiconductor electronics (DOI:10.1126/science.1257278).

Moreover, as mentioned by the Reviewer 3 in the comment (3) “*Stimuli-responsive micro/nanostructures are based on non-uniform responses upon external stimuli. Thus, this work will have a great potential in that field*”, we also think that nanostructures with out-of-plane shape variation (*i.e.*, non-uniform in size, shape, and/or composition along the axial direction) are capable of generating non-uniform responses as applying external stimuli, thus owning “potential applications in stimuli-responsive systems”.

All the above-mentioned points demonstrate the high application potentials of the out-of-plane controllable nanostructures. However, considering the space limitation of a paper, here we took the optical application as an example to show the advantages of out-of-plane well-defined nanostructures. In the future, we will definitely focus on the application of these out-of-plane controllable nanostructures in different fields.

To Reviewer 3:

Comments:

Yong Lei and his co-workers report a method to prepare complex yet high-fidelity nanoporous AAO templates by heterogeneous aluminium anodization rates with varied anodization voltages. Authors clearly introduced previous challenges and motivation for this work and demonstrated experimental evidence for their claim. As cited in ref. [22], authors previously reported a large-area nanofabrication of binary pored AAO templates in 2017. While there have been a number of applications of nanostructured AAO templates, reports on new strategy for nanofabrication has been limited. Hence, I enjoyed the reading this manuscript. In particular, I am impressed by the mix-and-match of different cross-sectional geometry for nanopores along the z-direction as well as spatial arrangements in x-y plane. Obviously, these new fabrication capabilities will contribute to overcome previous limitations and may open new horizons in various fields. Also, authors already performed extensive parameter studies and the experimental results are clearly supporting the main claim. Hence, I am favorable for publication of this manuscript in Nature Communications.

Answer: Many thanks for the nice comments.

1) However, considering free-form 3D nanoarchitectures produced by Nanoscribe, I think that “all-in-one platform” in the Abstract is overselling and ask authors to tone-down.

Answer: Thanks for the suggestion. We have revised it on page 1 of the revised manuscript:

“In light of the broad selectivity and high controllability, designable templates will provide a useful platform for well-defined nanostructuring.”

2) While it's for microstructures and based-on a growth mechanism, below reference is relevant and worthwhile to cite in the Introduction as an effort to prepare uneven cross-sectional shapes for small-scale fabrication. <https://www.nature.com/articles/ncomms7584>

Answer: Thanks a lot for your kind advice. We have cited this important article as Ref. 12 and properly mentioned in the introduction on page 2 of the revised manuscript:

“Various nanostructuring techniques have been developed, such as photo/electron-beam lithography, self-assembly, nanoimprinting, and template-based techniques¹¹ as well as material growth controlling¹², while almost none of these techniques fulfills all the above six capabilities of well-defined”

3) Stimuli-responsive micro/nanostructures are based-on non-uniform responses upon external stimuli. Thus, this work will have a great potential in that field. Below reference is worthwhile to cite in the Introduction and mention potential applications in stimuli-responsive systems. <https://onlinelibrary.wiley.com/doi/epdf/10.1002/pol.20210311>

Answer: We highly appreciate this insightful comment and appropriately cited this article as Ref. 7 on page 2 of the revised manuscript:

“Due to the compelling requirement of device miniaturization, synthesis of nanoscopic structures and their macroscopic integration into a large-scale array are fundamental to modern and future devices in the fields of optics¹, electronics², telecommunication³, biology⁴, energy conversion/storage^{5,6}, and stimuli-responsive materials⁷, etc.”

Potential application of well-defined nanostructuring in stimuli-responsive system is highlight on pages 12-13 of the revised manuscript:

“These well-defined nanostructures with high freedom of structural designability could result in some unique properties. Furthermore, external stimuli (*e.g.*, capillary force, light, magnetic field, and heat) will further adjust these properties in a dynamic way.⁷”

REVIEWERS' COMMENTS

Reviewer #1 (Remarks to the Author):

The requests for changes have been adequately addressed in the revised manuscript so that it can be published as is.

Reviewer #2 (Remarks to the Author):

The authors have clearly and comprehensively addressed the concerns from reviewers in the revised manuscript, which therefore can be accepted for publishing as it is.

Reviewer #3 (Remarks to the Author):

The authors addressed all the previous concerns and further improved the quality. So, I recommend for publication of this manuscript in Nature Communications.